# How Preemption Can Lead to Inequity

**DOI:** 10.3390/ijerph191710476

**Published:** 2022-08-23

**Authors:** Y. Tony Yang, Carla J. Berg

**Affiliations:** 1Center for Health Policy and Media Engagement, School of Nursing, George Washington University, 1919 Pennsylvania Ave. #500, Washington, DC 20006, USA; 2Department of Health Policy and Management, Milken Institute School of Public Health, George Washington University, 950 New Hampshire Avenue, Washington, DC 20052, USA; 3George Washington Cancer Center, George Washington University, 800 22nd Street NW, #7000C, Washington, DC 20052, USA; 4Department of Prevention and Community Health, Milken Institute School of Public Health, George Washington University, 950 New Hampshire Avenue, Washington, DC 20052, USA

**Keywords:** preemption, policy, law, covid-19, health disparities, social determinants

## Abstract

American cities and localities have historically been places of innovation and incubation when it comes to advancing equity and inclusion. Now, local governments in many states are leading the fight for stronger public health protections against COVID-19—through mask mandates, stay-at-home orders, and paid leave provisions, among other actions. However, state lawmakers have long used preemption—state laws that block, override, or limit local ordinances—to stifle local government action, often under pressure from corporate interests and political ideology. Through preemption, state lawmakers have obstructed local communities—often majority-minority communities—from responding to the expressed needs and values of their residents through policies. In this article, we first look at the context behind preemption and its disparate effects. After establishing a conceptual framework for measuring disparities, we discuss how the current COVID-19 pandemic is disproportionately harming the same communities that have been preempted from taking local action, limiting their ability to effectively combat the public health crisis. We argue that all stakeholders interested in health equity have a role to play in addressing the misuse of state preemption.

## 1. Introduction

Preemption is a legal doctrine that allows a higher level of government to “limit or eliminate the power of a lower level of government to regulate a specific issue” [1]. Preemption is neither inherently good nor bad, but it can function as a barrier to public health measures and exacerbate inequities [2]. Local governments are more responsive to the needs of their communities, and they also function as “laboratories” that experiment with public health policy to identify innovative solutions. Therefore, preemption that restricts the authority of local governments can have a negative impact on grassroots movements and public health innovation [2]. As such, preemption can stand in the way of effective public health measures. Likewise, local governments may be better suited to address the unique needs of vulnerable and disadvantaged populations in their communities by ensuring their representation and voice in making policy decisions. Furthermore, when state laws are used to restrict local authority to regulate certain areas, it can be especially harmful to racial/ethnic minorities and socioeconomically disadvantaged groups. One example of such implications is demonstrated through governmental action during the COVID-19 pandemic. Based on these experiences and the state of the literature, there is a dire need to study and better understand the implications of preemption on public health and health equity. Here, we will discuss how preemption can undermine health equity, describe potential frameworks and strategies that can guide decision-making, and take a closer look at how preemption plays a role in emerging national emergencies such as the COVID-19 pandemic.

## 2. Background

When used strategically, preemptive policies can serve as a powerful tool to ensure uniform regulation to protect against conflicts between different levels of government, as well as to advance wellbeing and equity [2]. However, most often, preemptive interference can impede local innovation, exacerbate inequities, and cause confusion and distrust between government levels [3].

Local government has the advantage of a more concerted, intimate level of governance, with governing officials often having first-hand experience with the challenges faced by community members. As a result, local policies are more directed, and the specificity with which the guidelines are developed can lead to unconventional, and at times revolutionary, policies that may serve as an example for broader legislation if deemed effective. Preemption eliminates this ability to utilize lower-level governments as “incubators for innovative policies” [4] by removing local key actors from the conversation and limiting the number of voices engaged.

While the impedance of such public health innovation can prevent local jurisdictions from addressing disparities in health, laws that restrict or prohibit local regulation can also prevent state and local governments from acting to protect their citizens [2], which often disproportionately impacts certain groups of people and exacerbates existing disparities [2]. Public health does not exist in isolation, and any policy that affects housing, internet access, or even the general economy, can lead to socioeconomic barriers to health for whole groups of people. For this reason, preemption of any kind quickly becomes a public health issue.

Moreover, preemption can create tension between different levels of government, pitting regulation agencies against one another, leading to confusion and public distrust [1]. Lower-level governments may push back on preemptive laws and attempt to regulate a restricted area or refuse to enforce preemptive orders. In times when cooperation, local innovation, and unified messaging would benefit public health, different levels of government may undermine one another’s authority because of the power struggle of preemption.

State legislatures have become more aggressive in their use of preemption in the recent past, as shown by a state-by-state analysis of preemption on the following seven policy areas: minimum wage (28 states), paid leave (23 states), anti-discrimination (3 states), ride sharing (41 states), home sharing (5 states), municipal broadband (20 states), and tax and expenditure limitations (42 states) [5]. For example, 23 state legislatures have passed laws that preempt the ability of municipalities to pass laws mandating employers within their jurisdictions provide paid leave [6]. When states preempt municipalities’ authority to pass paid sick and family and medical leave laws, they are not only limiting local control but also hurting the overall health and wellbeing of employees. In a study, researchers found that 68 percent of those without paid sick leave went to work with a contagious illness [7]. With more sick people at work, there is a greater likelihood of others becoming ill, thereby decreasing overall productivity and wellbeing.

Also, with reproductive rights now in states’ hands in the aftermath of the Supreme Court’s overturning of Roe v. Wade, some local governments are challenging state statutes’ power and taking steps to protect abortion [8]. However, if the state has preempted the regulation of abortion, municipalities are not authorized to adopt an ordinance regulating abortion, further compounding racial disparities in maternal health [9].

## 3. Disparate Effects

When policy causes harm, it rarely does so in a uniform fashion across all members of society. Accordingly, preemption can have a discriminatory effect on racial/ethnic minorities, socioeconomically disadvantaged, and other vulnerable groups [3]. The effects of preemption touch on public health, economic opportunity, voting rights, civil rights, and racial equality. When preemption affects local regulation of economic interests, housing, and other areas of life, the fallout of these policies can have greater impacts on public health. For example, many states preempt local governments from improving broadband internet access [10], which affects the way people connect to the world, including their access to telemedicine, education, and remote work [11,12]. While technology has increased healthcare accessibility greatly during the pandemic, allowing individuals to attend a virtual visit during lunch, or work from home on the day of an appointment, policies that restrict a government’s ability to ensure equitable internet access only widen gaps of health disparity.

Additionally, preemptive restrictions limiting local and fiscal authority to raise and spend revenue prohibit localities from using such revenue to reinvest in local services and the local economy [1]. Other preemptive policies have been used to restrict housing regulations, including zoning, rent, and anti-discrimination policies. These laws restrict the housing security and rights of vulnerable people. Additionally, states may preempt the ability of local jurisdictions to mandate earned sick days or other employee benefits, limiting workers’ ability to obtain medical care [13]. Some preemptive laws also limit civil rights protections for groups, such as sexual minorities, preventing local governments from protecting these individuals from discrimination [13].

Preemption can also be imposed to favor certain groups over others. In the South, preemption has a long history of reinforcing racism and favoring white property owners [3]. Southern state governments have long used preemption to create barriers for people of color. Preemption may also favor the interests of corporations or certain industries that have economic interests in deregulation. In these cases, preemption restricts the power of local governments to address the needs of their vulnerable constituents and perpetuates disparities among racial and socioeconomic groups.

The structure and framework of state and local laws also influence how preemption can be used to disadvantage particular groups of people. “Home rule” is the legal structure governing what powers a local government may exercise and how states can limit that authority [4]. Some states have home rule laws known as “Dillon’s Rule,” which restricts local authority to powers expressly granted by the state [3]. Dillon’s Rule creates substantial barriers to local government action, creating a presumption of state preemption. The preemptive authority of Dillon’s Rule strips local power and disproportionately harms “black and brown workers, low-income workers, and women” [3].

Preemption can be used as a tool to stymie local power to regulate labor standards, civil rights, public health and safety, technology, environmental protection, land use, and taxes [14,15,16,17,18,19,20]. Laws regarding these issues both directly and indirectly impact health, safety, and economic opportunities—particularly for racial minorities and low-income groups. Local governments should be given authority to act in the interests of the most vulnerable members of their communities.

## 4. Conceptual Framework for Measuring Disparities

One such attempt at a structured, systematic evaluation of the effects of preemptive policy is the Racial Equity and Policy Framework (REAP) [21]. While REAP focuses on racial equity and inequity, the spirit of this framework is applicable to communities and populations disproportionately impacted by various health issues and related policies. REAP prompts policymakers and stakeholders to evaluate a policy according to a number of key assessments. At the center of these assessments are three considerations: disproportionality, decentralization, and voice (see Table 1). Collectively, these pillars are examined to evaluate how a policy distributes benefits and burdens, through which level of government the policy is implemented, and the position that people of color—or other disadvantaged populations—are provided to influence and play a role in shaping policy.

An important note of this assessment is that it does not just focus on the communities the policy is intended to impact and the potential outcomes of a policy. Rather, the framework includes a closer look at the actors and institutions responsible for drafting the policy, examining the biases that may exist, and the context through which the policy will be constructed and implemented. In this way, the REAP framework not only identifies and measures the inequitable outcomes of a particular policy but also begins to unravel the components of policy construction that may have led to inequities. Ultimately, the goal of this assessment is not to label a policy as racist but rather to examine the effects of a policy in such a way that allows policymakers, stakeholders, and academics to consider the avenues through which a policy may enable racial inequity. In doing so, key actors will be able to use this information to guide future policy making and to construct policies designed to combat systemic and structural inequities.

## 5. Preemption during COVID-19

One way to consider the REAP framework and its utility is to examine circumstances in which there were limitations to how policies were implemented, particularly policies that likely have implications for health inequities. Preemption during COVID-19 provides several good examples. In the case of this interstate and international public health emergency, preemptive actions were taken to ensure a uniform response across federal, state, and local jurisdictions [4]. For example, Congress enacted the Public Readiness and Emergency Preparedness Act which preempted some state and local laws during the pandemic [1]. This action helped pharmacists order and administer COVID-19 tests and vaccinations, broadly increasing the availability of these services.

Conversely, state preemption of local measures to prevent the spread of COVID-19 may amplify health inequities [1]. In response to the COVID-19 pandemic, states used their preemptive power to limit the authority of local governments in many ways and in doing so, blocked local action that was necessary to protect public health [3]. Many states established regulatory ceilings to prevent local governments from imposing stricter requirements than the state [3]. Some states exercised this authority by restricting local governments from imposing stricter or less strict social distancing regulations and business shutdowns [3]. Other states used preemption to regulate shutdown, masking, and curfew orders [1]. A few states created a regulatory vacuum by not issuing state laws (i.e., not setting a floor or ceiling) but restricting the authority of local officials to regulate in certain areas [1].

Depending on the nature of the state versus local policy context, state preemption can lead to the exclusion of racial/ethnic minorities and low-income people from opportunities and health benefits that local laws could provide by limiting their voice and involvement in policy considerations [1]. For example, local authorities may want to implement mask mandates, improve broadband access, or offer sick leave policies—particularly if local authorities recognize the voices of those disproportionately impacted by the lack of such actions. As another example, housing became a critical issue during COVID-19. As employment rates plummeted, many families found themselves forced out of their rental units with no means to pay [1]. Despite growing concern over housing stability, states with previously established preemption laws, such as Florida and Illinois, effectively blocked local governments from regulating rent during the pandemic [22]. If local governments are preempted from acting in these areas to protect their constituents, it is likely that there will be short- and long-term implications for health inequities, particularly for racial minorities and lower socioeconomic groups.

Unfortunately, most pandemic-era examples of preemption did not promote public health. Instead, they reduced the health and safety protections that state and local governments could impose [3]. This often left people who were already in vulnerable situations worse off when their local governments could not step in to enact policies or provide aid. For example, the Mississippi Governor’s statewide mandatory stay-at-home order preempted localities from enforcing stricter restrictions, forcing its municipalities to rescind local stricter measures already in place [23]. When Texas Governor let his state’s stay-at-home order expire, he made it clear that his new order “supersedes all local orders”; Texas Attorney General then warned officials in Austin, Dallas, and San Antonio to roll back “unlawful” local emergency orders that imposed local public health safety measures, hinting that there would be lawsuits if they did not comply [24].

Effectively, the pandemic “deepen[ed] existing inequalities along racial lines”, and preemption played a key role in this outcome [3]. Preemption was used to restrict local authority to mitigate public health and the economic fallout of the pandemic. By setting restrictions on what actions localities could take to address community needs, preemptive laws served as a tool to reinforce inequalities among vulnerable groups during COVID-19. Preemption policies that preexisted the pandemic—especially those concerning broadband access, discrimination, and business regulation—created new challenges for local governments trying to respond to the needs of their communities [3]. Moreover, this stalled local action and prevented the development of innovative solutions at the local level [3] during an unprecedented time when innovative and creative problem-solving was necessary to promote the health and safety of local communities. Finally, states blocking local efforts to respond to the pandemic created friction between the different levels of government, which undermined their capacity to work together to advance health equity [3].

## 6. Conclusions: Addressing Preemption and Its Role in Health Equity

As noted, preemption does not inherently prevent positive public health outcomes. In fact, scholars suggest that preemption could have been used to promote health and economic equality by advancing public health goals and ensuring that local governments were better equipped to deal with the pandemic [4]. However, because preemption is being used as a tool to perpetuate inequities and reduce innovation, federal and state legislators should “avoid framing preemptive legislation in a way that hinders public health action [13]”. Moving forward through the COVID-19 pandemic and beyond, policymakers should consider how preemption can be used and adapted to better serve the interests of all people, promote public health, and reduce inequities.

There is a need for fresh approaches to preemption that elevate social, economic, and health interests and limit inequity and discrimination [25]. Experts recommend many steps that federal, state, and local governments can take to improve preemption as it relates to public health and equity (see Table 2). Each of these recommendations attempts to place community equity at the center of policy implementation in what legislatures call an “equity-first framework”.

The equity-first preemption framework could improve the effects of preemption on public health, equity, and good governance [25]. Some scholars characterize preemption as “both a cause of and a means to alleviate” inequities, and equity-first reconciles preemption’s potential to advance and hinder health equity [25]. This framework would assess when preemption will enhance or inhibit equity and ensure that local governments are not preempted from implementing policies that would benefit the health of their communities in a just way [25].

In order to ensure policies developed under the equity-first framework promote health equity, policymakers can utilize the strategy outlined in the REAP framework described above to examine the anticipated effects of the drafted guidelines. By imagining the ways in which a written policy could be enacted to exacerbate inequities, lawmakers can understand the components and context of the law that cause harm, offering an opportunity to reconstruct the policy without these implications. Table 1 highlights key aspects of policy assessment under the REAP framework that may be applied more broadly.

Federal, state, and local laws and policies play significant roles in perpetuating health inequities. Local governments are sometimes preempted from addressing the needs of their communities when preemption restricts their ability to regulate in certain areas. Local action is often tailored to the unique challenges of their individual communities, and state preemption bars the “pursuit of healthier, more equitable futures” [25]. This inhibits responsiveness and impedes efforts to remedy existing laws that may have discriminatory effects.

The equity-first framework also changes the way in which preemption can be categorized. Rather than viewing preemption through the lens of a floor, ceiling, or vacuum, this proposed framework would shift the analysis from the mechanics of the law to the anticipated impact on health and health equity.

In order to strike the proper balance between state and local levels of government, an equity-first preemption framework can harness the power of preemption to counter harmful policies and promote fairness. Advocates for equity-first frameworks say that while this approach is promising, more research needs to be done in this area to establish an effective framework based on empirical evidence. If equity-first is pursued as a reform to the current mechanical preemption framework, research would be essential to developing an effective model.

## Figures and Tables

**Table 1 ijerph-19-10476-t001:** Conceptual Framework–A Health Equity Framework (adapted from the REAP Framework).

	Decentralization	Disproportionality	Voice
**Actors, Institutions and Networks**	Are key actors and institutions operating at the national, state, and/or local level? How do these levels of government connect to other actors and institutions across other levels?	Are key actors from, representatives for, or connected in network to, disproportionately impacted communities?	Do they meaningfully engage and incorporate disproportionately impacted communities, and center their interests in the policy process? Whose voices are most powerfully connected across and within networks?
**Contexts and Events**	What are the economic and political contexts within which policy is being enacted and implemented and what are the relevant policies and events?	How are economic and political contexts, policies, and events impacting communities disproportionately?	What roles in shaping these contexts are held by disproportionately impacted communities and how do these policies and events disproportionately affect these communities?
**Ideas**	What ideas are reflected in policy outputs and discourse, and how do these vary at the national, state, and/or local levels?	How are disproportionately impacted communities constructed or depicted in policy ideas?	What roles in shaping policy discourse and ideas are held by disproportionately impacted communities?

Adapted from: Michener, A Racial Equity Framework for Assessing Health Policy (Commonwealth Fund, January 2022). https://doi.org/10.26099/ej0b-6g71 (accessed on 15 August 2022).

**Table 2 ijerph-19-10476-t002:** Recommended steps to improve the effects of preemption on public health and equity.

** *Provide a foundation from which to evaluate impact of preemption* **
Invest in research to gather empirical data on the public health effects of preemption [2]
Develop a robust evidence base regarding preemption (from various content areas and diverse approaches) to inform preemption policy considerations
Include savings clauses (explicit statement that the law does not preempt lower levels of government from enacting stronger legislation to protect public health)
** *Establish best practices for preventing exacerbations of health inequities* **
Emphasize that preemptive clauses in legislation should draw on established evidence base
Ensure that core to all policies is consideration of those most likely to be disproportionately impacted by the policy
Involve representation of those most disproportionately impacted in all phases of policy drafting
Obtain input from the public health science community to determine whether preemption could have positive or negative public health benefits
** *Policy considerations* **
Do not enact preemptive public health laws that are not supported by scientific evidence
Consider providing waiver provisions in preemptive laws
Remove existing state preemption of more protective local laws related to COVID-19
Strengthen the “home rule” to promote local authority to regulate in their own communities
Consider federal preemptive intervention to combat the misuse of state preemption
Enact preemptive legislation that serves as a regulatory floor rather than a regulatory ceiling to reserve the power of local authorities to enact stricter laws
Disallow regulatory vacuums

## Data Availability

Not applicable.

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
