# Peer review of "How Preemption Can Lead to Inequity"

_ijerph, 2022, doi:10.3390/ijerph191710476_

Round 1

Reviewer 1 Report

Dear Authors

I red carefully your paper untilted "How preemption can lead to inequity".

This topic is relevant for public health prevention.

To my opinion, your paper is more a position paper. It is difficult to follow your goals, and what are they? it is not suitable for a scientific peer-reviw journal such as IJERPH.

introduction, background are sometimes repetitiv. the part on COVID-19 very few .

good luck for future 

Author Response

  1. We have added a sentence at the bottom of our intro to establish our goals (lines 44-47).
  2. We have expanded our Covid-19 discussion (lines 201-207)

Reviewer 2 Report

This is an interesting argument and certainly worthy of consideration.  However, it seems to lack two key elements that would certainly strengthen its argument.  First, the article doesn't present any data about whether or not preemption at the local level is really a problem.  The article makes the case that in CAN happen, but how often has it happened, and what were the effects?  The article talks about the REAP evaluation process.  Has this process been implemented so that the public health community has data related to the evaluation process that can be shared?  My point is that why publish an article decrying a problem that doesn't exist? 

Second, the article would be much compelling with a specific example of how preemption actually played out related to COVID to the detriment of a local community.  What did a local community want to do that it couldn't do due to preemption?  This would not necessarily verify a that preemption is significant problem (one example would not be sufficient to establish a systematic, national problem) but it would clarify to readers how preemption actually works, and illustrate its consequences.  

If the authors cannot provide data about the nature of this problem, and cannot provide a specific example, then this article does not appear worthy of publication.  If they can, then it would be an important article to publish.  

Author Response

  1. We have added more info about preemption prevalence data, increased uses in the recent years and examples (lines 77-93).  
  2. We have provided two examples of Covid preemption. (lines 201-207).

Reviewer 3 Report

The commentary "How Preemption Can Lead to Inequity" discusses the implications of states using preemption to limit local governance and decision making, often resulting in inequity and exclusion. The scope of preemption in this manuscript focuses primarily on the COVID-19 pandemic with the author presenting a potential equity-first preemption framework to help strike a balance between state and local governments. 

Overall, the commentary is clearly articulated, with minimal grammatical errors. The only significant suggestion is potentially connecting this commentary to other current public health concerns where issues with preemption is evident, such as with the current increase in abortion restrictions and bans in several states (following the overturning of Roe v. Wade). 

Author Response

We have added a paragraph to discuss preemption and in aftermath of Supreme Court’s overturning of Roe v. Wade. (Lines 89-93)

Round 2

Reviewer 1 Report

dear authors,

You reply to my comments 

Best regards

Reviewer 2 Report

The authors have addressed my concerns indicating that the article is now worthy of publication.